# UV Light-Generated Superhydrophilicity of a Titanium Surface Enhances the Transfer, Diffusion and Adsorption of Osteogenic Factors from a Collagen Sponge

**DOI:** 10.3390/ijms22136811

**Published:** 2021-06-24

**Authors:** Masako Tabuchi, Kosuke Hamajima, Miyuki Tanaka, Takeo Sekiya, Makoto Hirota, Takahiro Ogawa

**Affiliations:** 1Department of Orthodontics, School of Dentistry, Aichi-Gakuin University, 1-100 Kusumoto-cho, Chikusa-ku, Nagoya 464-8650, Aichi, Japan; machako@dpc.agu.ac.jp (M.T.); hamajima.k0329@gmail.com (K.H.); minkey646@gmail.com (M.T.); tsekiya@dpc.agu.ac.jp (T.S.); 2Weintraub Center for Reconstructive Biotechnology, Division of Advanced Prosthodontics, UCLA School of Dentistry, Los Angeles, CA 90095-1668, USA; togawa@dentistry.ucla.edu; 3Department of Oral and Maxillofacial Surgery/Orthodontics, Yokohama City University Medical Center, 4-57 Urafune-cho, Minami-ku, Yokohama 232-0024, Kanagawa, Japan

**Keywords:** UV treatment, hydrophilicity, cell recruitment, atelocollagen sponge

## Abstract

It is a significant challenge for a titanium implant, which is a bio-inert material, to recruit osteogenic factors, such as osteoblasts, proteins and blood effectively when these are contained in a biomaterial. The objective of this study was to examine the effect of ultraviolet (UV)-treatment of titanium on surface wettability and the recruitment of osteogenic factors when they are contained in an atelocollagen sponge. UV treatment of a dental implant made of commercially pure titanium was performed with UV-light for 12 min immediately prior to the experiments. Superhydrophilicity on dental implant surfaces was generated with UV-treatment. The collagen sponge containing blood, osteoblasts, or albumin was directly placed on the dental implant. Untreated implants absorbed only a little blood from the collagen sponge, while the UV-treated implants absorbed blood rapidly and allowed it to spread widely, almost over the entire implant surface. Blood coverage was 3.5 times greater for the UV-treated implants (*p* < 0.001). Only 6% of the osteoblasts transferred from the collagen sponge to the untreated implants, whereas 16% of the osteoblasts transferred to the UV-treated implants (*p* < 0.001). In addition, a weight ratio between transferred albumin on the implant and measured albumin adsorbed on the implant was 17.3% in untreated implants and 38.5% in UV-treated implants (*p* < 0.05). These results indicated that UV treatment converts a titanium surface into a superhydrophilic and bio-active material, which could recruite osteogenic factors even when they were contained in a collagen sponge. The transfer and subsequent diffusion and adsorption efficacy of UV-treated titanium surfaces could be useful for bone formation when titanium surfaces and osteogenic factors are intervened with a biomaterial.

## 1. Introduction

Dental implant therapy has been expanded with the application of bone augmentation techniques and socket preservation procedures [1,2,3]. However, those procedures can be a risk factor for reducing the success rate of implant treatments [4]. Cells, growth factors and scaffolds are important elements for bone augmentation [5]. In addition, cell-material interactions, i.e., the titanium-osteogenic cell interaction, are essential to achieve osseointegration, while material-material interactions, i.e., the interaction of titanium with other materials seem to play a key role in developing consistent integration between a titanium surface and the developed/augmented bone.

If only bone formation is required, it is sufficient to use scaffold materials for bone formation in the area when needed. However, when implants and scaffold materials are used together, the osteogenic factors may move to only the scaffold without attaching to the implant, resulting in a decrease in the supply of those factors around the implant body. Therefore, the implant body is often placed several months after bone augmentation. In order to solve these problems, the implant body needs to attract blood at least as much as the scaffold material.

Among currently available scaffold materials, Type I collagen is commonly used because of its biocompatibility and in vivo degradability [6]. Type I collagen consists of 95% helix and 5% telopeptide types. The telopeptide portion is removed because of its high antigenicity, and is applied as atelocollagen, which is a natural polymer material with low antigenicity [7]. Atelocollagen fibers are stable in vivo, but cell migration is low. Heat-denatured collagen (gelatin) has excellent cell migration properties but has low physical strength stability. Hence, they are mixed and used together so that the advantages of both can be harnessed [8]. In addition, Type I collagen has been reported to play an important role in angiogenesis [9] and osteoblast differentiation by signal transduction via integrins [10]. Thus, an atelocollagen sponge is an excellent bone replacement material because it stores blood containing cells and growth factors [11]. The atelocollagen sponge, Teruplug^®^, has been developed to enhance extraction socket healing by supporting blood clot formation in the socket and attracting newly formed bone arising from the socket wall [12]. Mesenchymal stem cells (MSC) existing oral tissue is also adsorbed to the materials. Oral MSCs plays an important role of not only tissue regeneration but local immunomodulation [13].

Ultraviolet (UV) treatment of titanium implants is a method that improves osseointegration between the titanium implant and bone tissue [14,15,16,17,18,19,20,21,22,23,24,25,26,27,28,29,30,31,32]. Hydrocarbons adhere to the surface of titanium over time, resulting in compromised osteoblast attachment and osseointegration [33,34,35,36,37,38]. UV light removes hydrocarbons from titanium surfaces and exposes Ti^4+^ sites, which promotes the interaction between cells and surfaces [39,40,41,42,43,44]. Moreover, UV treatment changes the surface wettability and converts implant surfaces from hydrophobic to hydrophilic [42,45,46,47,48,49]. Then, the UV-treated implant surface enhances an attachment and proliferation of osteogenic cells, resulting in enhanced osseointegration, which is demonstrated by a near 100% bone-implant contact in an animal model [34,40,50,51]. 

Here, we hypothesized that the interaction between UV-treated superhydrophilic titanium surfaces and aterocollagen sponges produces synergistic effects in recruitment of osteogenic factors to the titanium surfaces. Furthermore, blood could be retained around the implant at the same level as or higher than the blood absorption capacity of the aterocollagen sponge. We evaluated recruitment ability of UV-treated dental implant surfaces, which directly contacted with collagen sponges containing osteogenic factors.

## 2. Results

### 2.1. Changes in Wettability of Implants before and after UV-Treatment

A water droplet was formed on untreated implants that showed their hydrophobicity, whereas no water droplet was formed on the UV-treated implants, indicating their surface had superhydrophilicity (Figure 1A). The contact angle between a water droplet and the implant surface was 68.3° on average in the untreated group, while it was 0.0° in the UV-treated group. There was a significant difference between two groups (Figure 1B).

### 2.2. Transfer of Blood from the Collagen Sponge to the Implant

In untreated implants, the migration of blood from the collagen material to the implant body was limited to the contact area (Figure 2). When the collagen sponge was removed, transferred blood was confirmed only on the upper and side surfaces. On the contrary, it was clear that a large amount of blood transferred from the collagen sponge to the UV-treated implant, immediately and rapidly. The transferred blood distributed mostly through the entire implant surface along its groove without dropping downwards.

The amount of blood transferred from the collagen sponges to the untreated implants was 3.93 mg, whereas it was 11.5 mg for the UV-treated group, which showed that the amount of blood transferred from the collagen sponges was significantly increased by UV treatment (Figure 3A). The average ratio of the area, which blood distributed to the implant surface area was 18.6% in the untreated group, whereas it was 67.6% in the UV-treated group. Thus, the UV treatment significantly increased the implant area covered by blood migrating from the collagen material (Figure 3B).

### 2.3. Transfer of Osteoblasts from the Collagen Sponge to the Implant

When the collagen sponge with a culture medium containing osteoblasts was placed on the UV-treated implant, the solution spread and widely distributed around the implant surface., whereas mostly no solution spread on the untreated implant (Figure 4A). The number of viable osteoblasts transferred to UV-treated implants was significantly greater than that to untreated implants (Figure 4B). Then, the ratio of transferred osteoblasts was approximately 6% in the untreated group, whereas it was approximately 16% in the UV-treated group, respectively. The value of UV-treated implants was significantly and approximately three-times greater than that of untreated implants (Figure 4C).

### 2.4. Transfer of Albumin from the Collagen Sponge to the Implant

The rates of albumin adsorbed to the collagen sponge in the untreated and UV-treated groups were 22.2% and 21.5%, respectively (Figure 5A). An estimated rate of albumin tranferred from the collagen sponge to the implant was 7.4% in the untreated group and 12.6% in the UV-treated group (Figure 5B). The average weight of the transferred albumin on untreated and UV-treated implants was 0.022 g and 0.023 g, respectively. The weight ratio between the transferred albumin on the implant and measured albumin adsorbed to the implant was 17.3% in the untreated group and 38.5% in the UV-treated group. The weight ratio of UV-treated implants was significantly greater than that of untreated implants (Figure 5C).

## 3. Discussion

UV-treated implants showed excellent wettability in the present study. The wettability of titanium surfaces influenced osseointegration of dental implants [14,52,53,54,55,56]. Protein adsorption and cell attachment was interfered with hydrocarbon contamination or air bubbles caused by hydrophobic surfaces [30,57]. There have been previous reports on the hydrophilicity of implants after UV treatment [39,50]. The present results revealed that the contact angle between the implant and water droplet changed from about 60–70° to 0 ° after UV treatment. UV treatment removes hydrocarbons adhering to titanium surfaces and this generates its superhydrophilicity [39,50]. UV light generated superhydrophilic titanium surface is assumed to be oxygen vacancies at bridging sites of TiO_2_, resulting in appearance of Ti^3+^ site, on which dissociative water is easily adsorbed [39]. Before UV-treatment, TiO_2_ is covered with hydrocarbons, which inhibit water adsorption [39]. A superhydrophilic titanium surface has an advantage in attracting proteins and osteoblasts, which results in osteoblast attachment is enhanced [34,36,39,50,58,59]. Surface hydrophilicity/hemophilicity enables attraction of blood and osteogenic cells followed by protein adsorption, cell attachment and osseointegration. Electrostatic interaction of TiO_2_ surfaces play an important role of interaction with cells [58,60]. When the surface is negatively charged it needs some intermediation such as ion and protein to connect with osteoblast, whereas positively charged surface enables to directly connect with osteoblast [58]. Before UV-treatment, TiO_2_ surfaces are electronegative and attract proteins and cells with as aid of divalent cations, such as Ca^2+^ and proteins, while, after UV-treatment, TiO_2_ surfaces, which exposed TiO_2_, directly connect with osteoblast [58,60]. The present results revealed that the advantageous effect of UV-treatment on titanium surfaces was still active even though the collagen sponge was placed next to the titanium implant. Blood, osteoblasts and albumin adsorbed in the collagen sponges were transferred to UV-treated implants more efficiently than the untreated implants, and the difference was significant. The amount of blood and osteoblasts transferred to UV-treated implants was three-times greater than that to untreated implants. Both blood and osteoblasts-containing media were widely spread and distributed on the entire surface of UV-treated implants, whereas only a little part of the untreated implant adsorbed them. The results for albumin, which is a protein required for adhesion of osteoblasts [60], were different from the results for blood and osteoblasts. The present results for albumin adsorption showed that UV-treated implants tended to adsorb a greater amount of albumin rather than untreated implants although there was no significant difference. However, the weight ratio of the measured albumin adsorbed to UV-treated implants was significantly greater than that to untreated implants. The solution containing albumin was widely distributed on UV-treated implant surfaces because of its superhydrophilicity, while the solution remained at a limited part on untreated implant surfaces, resulting in the significant difference in measured albumin between untreated and UV-treated implant even though mostly the same amount of albumin was transferred on both implants. The adsorption of albumin on titanium implants differed depending on the conditions, such as surface pH [60] and surface properties at the micro level [36,61]. Nonetheless, UV-treated titanium implants adsorbed much more albumin than untreated titanium implants, suggesting that UV treatment contributes to the change in the amount of adsorbed albumin.

UV-treated titanium can recruit osteogenic cells from a distant bone surfaces [17,62]. That is to say, UV-treatment enables an implant to form bone tissue around itself if there is a gap between the implant surface and bone tissue [17,62]. However, since the capability has a limitation, it is assumed that UV-treated dental implants need much more bone tissue than its bone formation ability. Inefficient bone formation surrounding the peri-implant area causes bone resorption. An atelocollagen sponge used in the present study was developed to enhance extraction socket healing by supporting blood clot formation in the socket and attracting newly formed bone arising from socket wall [8], indicating a distant area apart from the dental implant placement area is effectively utilized. The atelocollagen sponge consists of not only Type I collagen but also Type III collagen [63], which is related to the bone healing process [64]. Type III collagen deposited in granulation tissue increases in the early stage of extraction socket healing [64], and appears at the remodeling site of human alveolar bone [12], suggesting it can be a suitable material for socket preservation. The placement of dental implants into a fresh extraction socket is often desired because rich bone formation can be expected to integrate to implant surfaces [65]. However, even though newly formed bone in the socket is utilized, the required bone formation and integration around implants could be insufficient, because the socket wall thickness is mostly incomplete. The present results revealed that UV-treated implants attract osteogenic factors from the collagen sponge without impaired clot formation in the sponge. Therefore, combination use of UV-treated implants with Teruplug^®^ could be effective for acquiring rich bone tissue around the implant when the implant placement site is anatomically compromised.

The presented results reveal that UV-treated titanium surfaces consistently recruited bloods, proteins and osteoblasts through the collagen sponge without impairing clot formation in the sponge, suggesting that the material-material interaction between superhydrophilic titanium surfaces and atelocollagen sponges was improved by UV treatment. This alternative interaction may induce synergistic effects on osteoblastic behavior in both materials. If superhydrophilic titanium surfaces and collagen sponges are successfully combined, it will be a powerful tool for transferring osteogenic factors to entire implant surfaces. These transfers and subsequent diffusion and adsorption efficacies of UV-treated titanium surfaces could be useful for bone formation when there is intervention of titanium surfaces and osteogenic factors by a biomaterial. Recently, Tatullo et al. [66] reported effectiveness of Oral MSCs to regenerative therapy, which can be transferred to dental implant surface though the materials such as collagen sponges. 

Longer observation period may reveal more clear difference between untreated and UV-treated implant. However, the transfer from the collagen sponge to the UV-treated implant was rapid and instantly reached to plateau compared to the untreated implant. Therefore, it was considered that the difference between untreated and UV-treated implant was the most conspicuous immediately after the collagen was placed on the implant. Then, osteoblastic behavior may be affected by material properties. Marrelli et al. [67] reported a standardized model using computer-aided design (CAD) technology can improve the evaluation of cell behavior on different biomaterials. Further investigations into osteoblastic behavior and bone formation processes are required to ascertain if the synergistic material-material interaction can be achieved in vivo. 

## 4. Materials and Methods

### 4.1. Preparation of an Atelocollagen Sponge and UV Light Treatment of Dental Implants

A bovine-derived atelocollagen sponge (Teruplug^®^, Olympus Terumo Biomaterial, Tokyo, Japan), which has a rocket shape, was used. The diameter of the columnar part of the collagen sponge was 8 mm. A part at the end of the collagen sponge was cut, and the remaining columnar part was cut to make a disc-shaped collagen sponge with a diameter of 8 mm and a thickness of 3 mm. The average weight of a disk-shaped collagen sponge was 0.0045 g (range; 0.0041–0.0053). Disk-shaped collagen sponges were used thoughout the study. Collagen sponges were immersed in the following material: blood, an osteoblast containing culture medium and an albumin containing liquid. After immersion, the collagen sponge was placed on a dental implant (Branemar, MarkIII TiUnite, RP, ø3.75 mm, Nobelbiocare, Goteborg, Sweden). As the collagen sponges are product for clinical use, dental implants, which are also used in practical treatment, were used to observe interactions between both medical products. Collagen sponges and dental implants were autoclaved before use. Furthermore, the dental implant had been stored for four weeks in a dark condition before use. Transfer activities of each experimental material from the collagen sponge to the dental implant was examined. UV-treatment of the implant was performed immediately prior to use with a UV-light device, Affiny^®^ (USHIO Inc., Tokyo, Japan) for 12 min. The device has UV-light source both inner walls. The distance between the walls was 21 mm and the implant was placed in the middle between the light sources. The distance between the light source and the implant surface was approximately 8.625 mm. The wavelength of the UV-light is equivalent to UVC rays. The protocols for animal experiments were approved by the University of California Los Angeles Animal Research Committee (approval number; 05-127; not dated) and were conducted in accordance with the United States Department of Agriculture Animal Experiment Guidelines.

### 4.2. Wettability of the Implant Body

The implant was placed parallel to the ground and a 5-µL of distilled water was dropped on the apex surface of the implant to evaluate the wettability of the implant with and without UV-treatment. The contact angle formed by the water droplet and the implant surface was measured using a contact angle meter (CA-X, Kyowa Interface Science, Tokyo, Japan). The angle formed by the line contacting the implant body and the upper surface of the water droplet was taken as the contact angle, and the untreated group and the UV-treated group were compared.

### 4.3. Transfer of Blood from the Collagen Sponge to Implant

Eight-week-old Sprague-Dawley rats (Charles River, San Diego, CA, USA) were sacrificed and blood was collected from the inferior vena cava. The blood was diluted with sodium heparin (Wako Pure Chemical Industries, Osaka, Japan) and was adjusted so that the final concentration was 10 µg /mL. A total of 3-mL of the blood was injected into a 6-cm polyethylene cell culture dish, and the collagen sponge was placed in the dish for 10 s. First, the weight of blood transferred to the implant was calculated by subtracting the weight of the implant before placing the collagen sponge from the weight of the implant after placing the collagen sponge for 3 min. Next, the collagen sponge containing blood was placed on the dental implant for 20 s. After removing the collagen material, the implant body was photographed from above, from both sides and from below, and then the weight of the implant body was measured. The area of blood on the implant surface was measured as a ratio of the whole implant surface using image analysis software Image J (NIH, Bethesda, ML, USA). The weight and area of blood transferred to the implant were compared between the untreated and UV-treated groups.

### 4.4. Transfer of Osteoblasts from the Collagen Sponge to the Implant

Following a previously established protocol [40], bone marrow cells isolated from the femur of 8-week-old male Sprague-Dawley rats were placed in alpha-modified Eagle’s medium supplemented with 15% fetal bovine serum, 50 μg/mL ascorbic acid, 10^−8^ M dexamethasone, 10 mM Na-ß-glycerophosphate and antibiotic-antimycotic solution containing 10,000 units/mL penicillin G sodium, 10,000 mg/mL streptomycin sulfate and 25 mg/mL amphotericin B. Cells were incubated in a humidified atmosphere of 5% CO_2_ at 37 °C. At 80% confluency, the cells were detached using 0.25% trypsin-1 mM EDTA-4Na and the density was adjusted to 5 × 10^5^ cells/cm^2^, and 100 µL of osteoblast-containing solution was soaked in the collagen sponge. An implant was placed in a well of a 12-well non-coated cell culture dish, and the collagen sponge with osteoblast-containing culture medium was placed on the implant for three minutes. The implant was fixed on the well with adhesive resin (Super-Bond C&B, Sun Medical Co, Ltd., Shiga, Japan) to avoid moving when the collagen sponge was placed. After three minutes, the collagen sponge was removed, the implant was moved to another unused well and the cells were detached and collected with 0.25% trypsin-1 mM EDTA-4Na. In addition, WST-1 assay (WST-1, Roche Appliec Science, Mannheim, Germany) was performed to detect viabilities in the osteoblasts adsorbed on the implant. The amount of formazan product was measured with a microplate reader at a wavelength of 450 nm. The detached osteoblasts were counted. Absorbances of WST-1 and the number of detached osteoblasts were compared between the untreated and UV-treated groups.

### 4.5. Transfer of Albumin from the Collagen Sponge to the Implant

A collagen sponge was immersed into 300 mL bovine serum albumin (BCA protein assay kit, ThermoFisher Scientific, Waltham, MS), which was diluted with saline to a concentration of 1 mg/mL in a 0.5-mL microtube (Eppendorf, Hamburg, Germany) for 10 min. An albumin-adsorbed collagen sponge was placed on a dental implant in the well of a 12-well non-coated cell culture dish for 40 s. After the collagen sponge was removed, the weight of the implant containing albumin solution was measured. Then, the implant was incubated for two hours in a humid environment at 37 °C. After incubation and gentle washing, the implant was placed in a 2-mL microtube with a 1 mL of phospahte buffer solution. Then, after pipetting, the tube was incubated for two hours in a humid environment. Bicinchoninic acid was added to a remnant of an albumin solution in a 0.5-mL microtube and an albumin-adsorbed implant in a 2-mL microtube, and the mixture was incubated at 37 °C for 1 h, then the contents were collected and the amount of albumin was quantified using a microplate reader at 562 nm. A calibration curve was prepared using an albumin solution with a concentration of 1 mg/mL and a phosphate buffer solution. The amount of albumin adsorbed to the collagen sponge was measured. An estimated amount of albumin tranferred from the collagen sponge to the implant was measured from the remnant of the albumin solution in a 0.5-mL microtube and the albumin adsorbed on the implant. In addition, a weight ratio between the transferred albumin on the implant and measured albumin was calculated. The rate of transferred albumin in the collagen sponge and rate of adsorbed albumin on the implant were compared between the untreated and UV-treated groups. Furthermore, based on the weight of the implant with the transferred albumin solution, the weight ratio of adsorbed albumin to transferred albumin on the implant was calculated and compared between the untreated and UV-treated groups.

### 4.6. Statistical Analysis

All of the experiments described above were performed in triplicate because the experiments were performed with normalized products which can be considered that every sample has identical properties. The average values between the untreated and UV-treated groups were compared. Data were expressed as the mean ± standard deviation. Statistical significance was evaluated using Student’s *t*-test at *p* < 0.05.

## 5. Conclusions

In this experiment, it was found that UV-treatment provides a favorable environment, which is a superhydrophilic surface, to attract osteogenic factors even when they are contained in materials placed next to the implant without impairing the original role of the collagen sponges, suggesting UV-treatment of implants further improves material-material interactions for bone formation even where it is difficult to achieve osseointegration. 

## Figures and Tables

**Figure 1 ijms-22-06811-f001:**
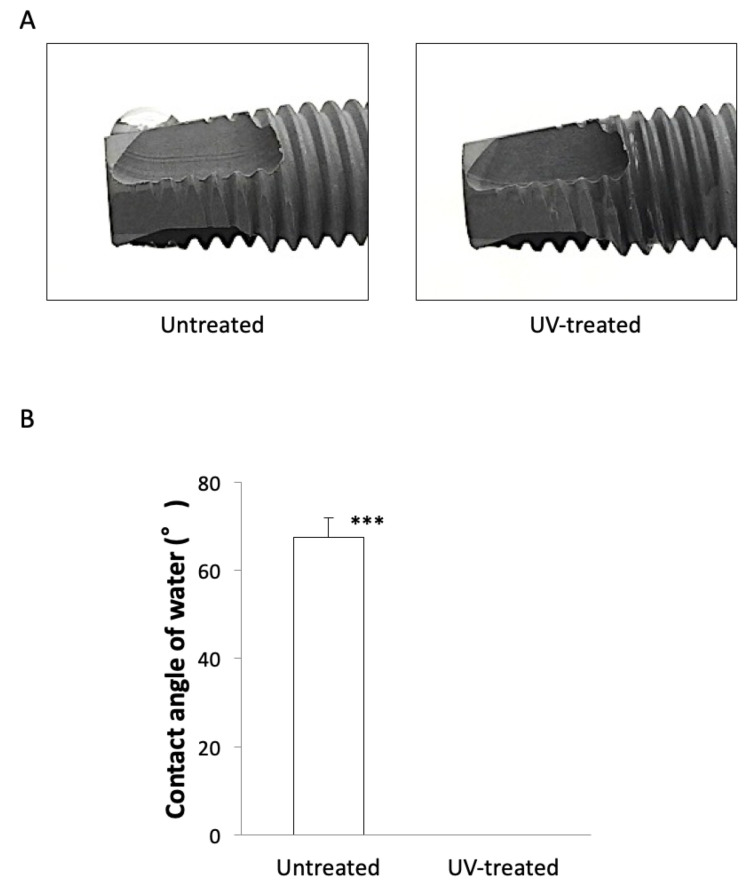
(**A**) Hydrophilicity of implants. The untreated implant (**left**) formed a water droplet, whereas the UV-treated implant (**right**) formed no water droplet, indicating its superhydrophilicity. Water spreads and distributes over the entire implant along its groove. (**B**) Contact angle between an implant surface and a water droplet. The contact angle on untreated implants was significantly higher than that on UV-treated implants. *** *p* < 0.001.

**Figure 2 ijms-22-06811-f002:**
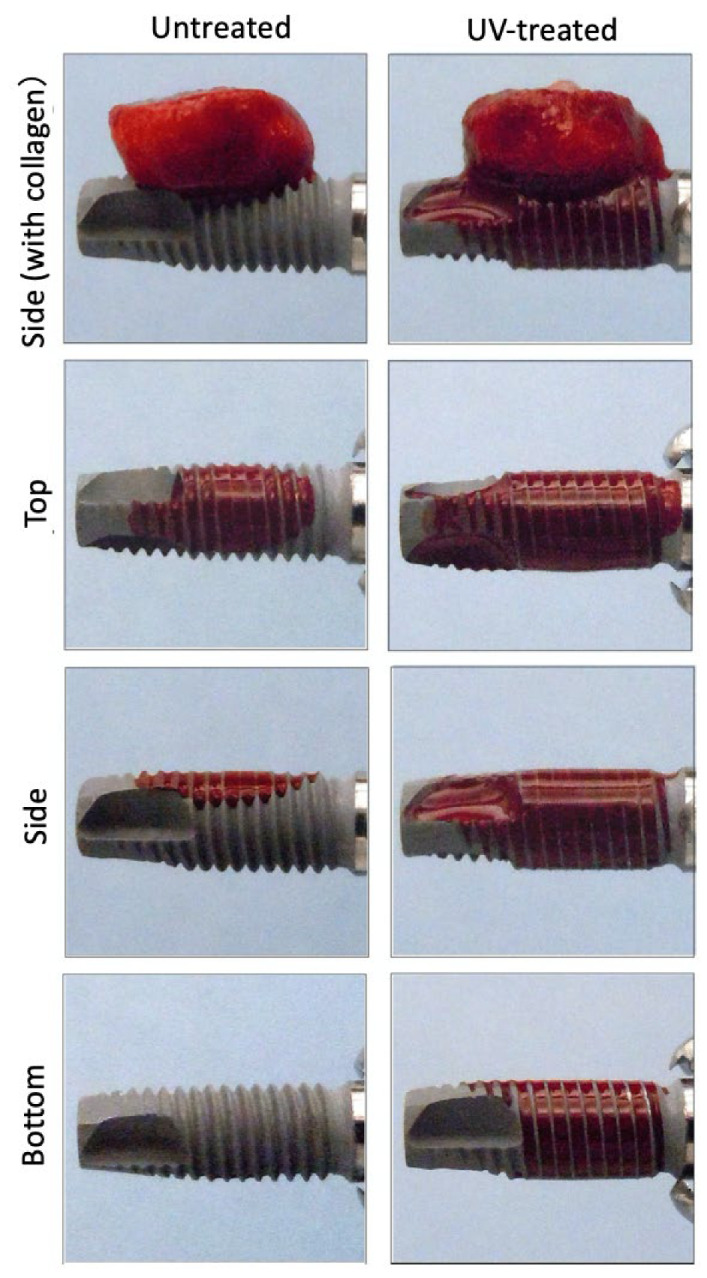
Transfer of blood from the collagen sponge to the implant three minutes after placing. Blood is retained in both untreated and UV-treated group. In the untreated implant, the transfer of blood was limited to the contact area, and in the UV -treated implant a large amount of blood moved from the collagen sponge to the implant. It could be confirmed that it moved throughout the entire implant. (**Top**); upper surface on which the collagen sponge was placed. (**Side**); side face of the implant. (**Bottom**); undersurface of the implant.

**Figure 3 ijms-22-06811-f003:**
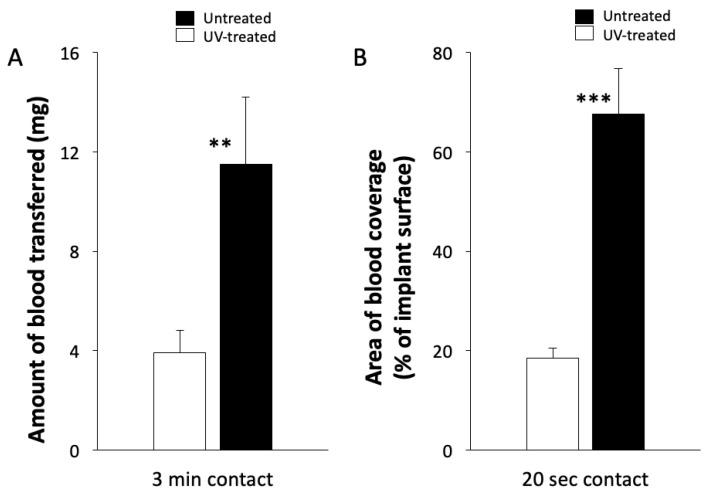
The amount of blood transferred to the implant after the collagen material with blood was left on the implant body for three minutes (**A**). Percentage of the area of the implant surface body covered by blood that transferred after 20 seconds contact (**B**). ** *p* < 0.01, *** *p* < 0.001.

**Figure 4 ijms-22-06811-f004:**
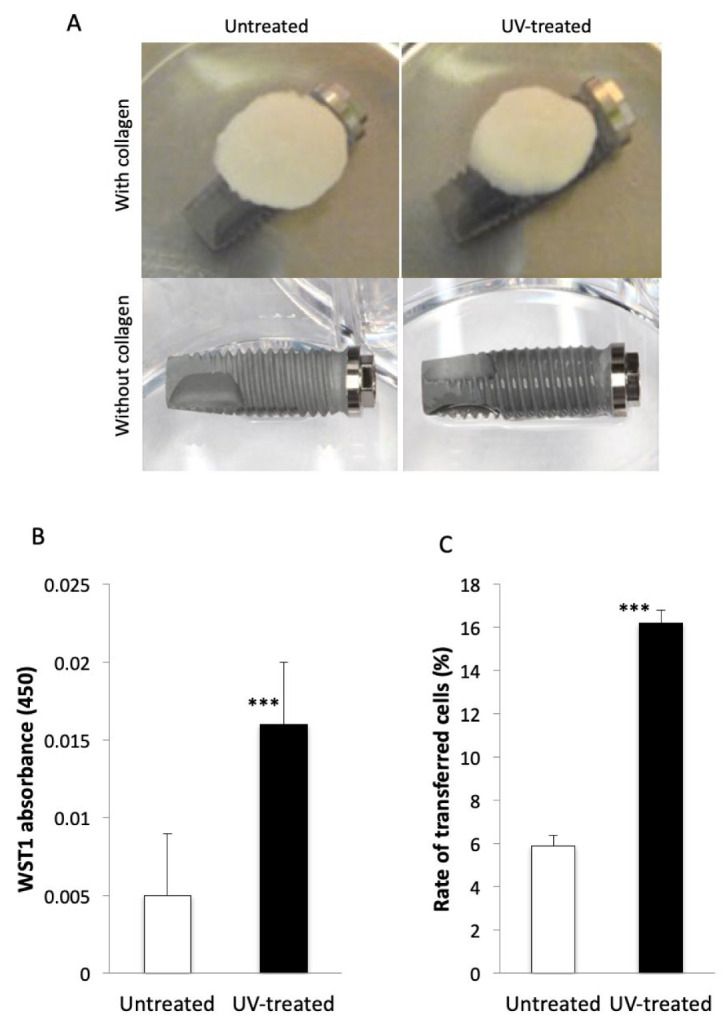
Transfer of osteoblasts from the collagen sponge to the implant. (**A**) A collagen sponge containing osteoblasts is placed on an untreated (**left**) and a UV-treated implant (**right**). After removal of collagen sponge, no solution is seen on the untreated implant (**left**), whereas osteoblasts-containing solution widely spreads and distributes on the UV-treated implant (**right**). (**B**) WST-1 assay reveals the number of viable osteoblasts transferred to UV-treated implants is significantly greater than that to untreated implants. (**C**) The rate of transferred osteoblasts from collagen sponges to implants is significantly greater in UV-treated group than untreated group. *** *p* < 0.001.

**Figure 5 ijms-22-06811-f005:**
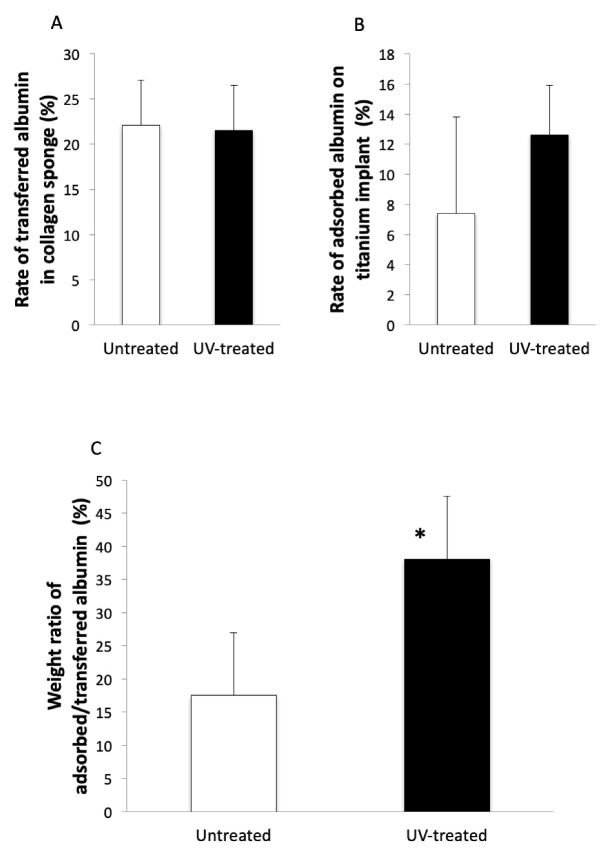
Transfer of albumin from the collagen sponge to the implant. Albumin was diluted with saline to a concentration of 1 mg/mL. (**A**) The rate of albumin adsorbed to the collagen sponge in the untreated and UV-treated groups was mostly equal. (**B**) An estimated rate for albumin tranferred from the collagen sponge to UV-treated implants was relatively higher than that transferred to untreated implants. (**C**) The weight ratio between the transferred albumin on the implant and measured albumin adsorbed to UV-treated imlants was significantly greater than that to untreated implants. * *p* < 0.05.

## Data Availability

The data that support the findings of this study are available from the corresponding author, Makoto Hirota, upon reasonable request.

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
