# Peer review of "UV Light-Generated Superhydrophilicity of a Titanium Surface Enhances the Transfer, Diffusion and Adsorption of Osteogenic Factors from a Collagen Sponge"

_ijms, 2021, doi:10.3390/ijms22136811_

Round 1

Reviewer 1 Report

The manuscript is very interesting and examines the effects of UV treatment of titanium surface on various aspects of osteogenic factors. Presented data show that UV treatment effectively enhanced the transfer, diffusion and adsorption of blood, osteoblasts and albumin from a collagen sponge.

Introduction and Discussion paragraphs are strength parts of the paper, which provides an overview of dental implant surgery. The authors finished the Introduction with interesting hypothesis but the last sentence in lines 80-82 with repetition “We investigated whether…..” or “….it was evaluated” makes the sentence difficult to understand. The authors also started the Discussion with the first sentence in lines 156-157 with similar repetition “In the present study, …….. in the present study” and later on in line 170 “…..much more than…..” and my suggestion is to use “     more efficient than……”. Also too much information about albumin from Results is repeated in lines 175-192 of Discussion. What type of proteins are mentioned in line 220 especially when authors examined only albumin and collagen type I? What means the abbreviation CAD in line 237? WSTs is a widely accepted form and also MSCs should be added to the Abbreviations.

Results paragraph is a weak part of the paper and need some corrections. Specific questions or comments are below:

  1. Why in Figure 3 description different time is used (for A 3 min but for B 20 sec)? and if it is important in line 119 “for twenty seconds” should be added.
  2. In Figure 4 description it should be added how osteoblasts were stimulated for mineralization and if they were washed from FBS proteins?
  3. In Figure 5 description it should be added what concentration of albumin was used?

Material & Methods paragraph is perfectly arranged and described in the style that other researchers may repeat experiments very easy.

Only 14 from 33 references are from years 2011-2021.

Author Response

To reviewer 1

Introduction and Discussion paragraphs are strength parts of the paper, which provides an overview of dental implant surgery. The authors finished the Introduction with interesting hypothesis but the last sentence in lines 80-82 with repetition “We investigated whether…..” or “….it was evaluated” makes the sentence difficult to understand.

Thanks for the comment. I have revised the sentence to “We evaluated recruitment ability of UV-treated dental implant surfaces, which directly contacted with collagen sponges containing osteogenic factors.”

The authors also started the Discussion with the first sentence in lines 156-157 with similar repetition “In the present study, …….. in the present study” and later on in line 170 “…..much more than…..” and my suggestion is to use “     more efficient than……”.

 Thanks for the comments. I have deleted the start words of “ In the present study” at the beginning of discussion and revised “much more than” to “more efficiently than”.

Also too much information about albumin from Results is repeated in lines 175-192 of Discussion.

Thanks for the comments. I have deleted the sentences between line 174 to 181(lines of original paper), which mentioned the reason why we investigated a weight ratio. I have tried to describe just a significant difference and its reason.

What type of proteins are mentioned in line 220 especially when authors examined only albumin and collagen type I? What means the abbreviation CAD in line 237? WSTs is a widely accepted form and also MSCs should be added to the Abbreviations.

Thanks for the comments. I have added “MSC” and “CAD” in the abbreviation.

Results paragraph is a weak part of the paper and need some corrections. Specific questions or comments are below:

  1. Why in Figure 3 description different time is used (for A 3 min but for B 20 sec)? and if it is important in line 119 “for twenty seconds” should be added.

Thanks for the comments. The transferred was completed around 20 seconds after the collagen sponge was placed on the implants. We have tried to describe that blood transferred to the implants in just 20 seconds, but still retained in the collagen sponges after three minutes. The rapid an instant transfer had been described in discussion section. So, I have added comments on figure 2 description for reader to understand that “Blood is retained in the collagen sponge after three minutes placing”.

  1. In Figure 4 description it should be added how osteoblasts were stimulated for mineralization and if they were washed from FBS proteins?

Thanks for the comments. The culture condition was described in method section. And, we used FBS for osteoblast cell culture but did not use for protein examination. The culture condition is usual method, which has described earlier (reference 14).

  1. In Figure 5 description it should be added what concentration of albumin was used?

Thanks for the comments. I have added the concentration in the figure description.

Reviewer 2 Report

Review comments of ijms- 1247014 manuscript

The current manuscript entitled “UV light-generated superhydrophilicity of a titanium surface enhances the transfer, diffusion, and adsorption of osteogenic factors from a collagen sponge” and authors have reported the experimental evidences in order to prove their findings with potential results. I would recommend the manuscript for publication after major revision and find the following comments.

  1. Authors have to address what type of functional groups is presented on the surface of the titanium dental implant material before and after UV-treatment.
  2. Authors have to explain what type of photochemical interactions occurred during UV treatment and simultaneously explain the photochemical mechanism of dental implant during UV treatment.
  3. In literature reported that surface activation of dental implants by UV-LED irradiation (Nagore. A.L et al, Int. J. Mol. Sci. 2021, 22, 2597). Current work is similar to the literature work, so that did not find any significance to the reported work as compared with reported work.

Author Response

To reviewer 2 

  1. Authors have to address what type of functional groups is presented on the surface of the titanium dental implant material before and after UV-treatment.

  1. Authors have to explain what type of photochemical interactions occurred during UV treatment and simultaneously explain the photochemical mechanism of dental implant during UV treatment.

Thanks for the comments. I have tried to describe a type of functional group on Ti surface and interaction/mechanism of cell attachment.

  1. In literature reported that surface activation of dental implants by UV-LED irradiation (Nagore. A.L et al, Int. J. Mol. Sci. 2021, 22, 2597). Current work is similar to the literature work, so that did not find any significance to the reported work as compared with reported work.

Thanks for the comments. They reported LED-based UVC removed carbon contamination on aged titanium surface and did not report wettability and blood and protein affinity which investigated in our report.

Round 2

Reviewer 2 Report

Review comments of revised ijms-1247014

The revised manuscript entitled “UV light-generated superhydrophilicity of a titanium surface enhances the transfer, diffusion, and adsorption of osteogenic factors from a collagen sponge” has been improved by the authors appropriate corrections. I strongly recommend this manuscript for publication without further revision.
